# Who feels like they belong? Personality and belonging in college

**Alexandria M. Stubblebine** [1], **Maithreyi Gopalan** [2]*, **Shannon T. Brady** [3]

**1** Independent Researcher, Ocala, Florida, United States of America, **2** Department of Education Policy Studies, The Pennsylvania State University, University Park, Pennsylvania, United States of America, **3** Department of Psychology, Wake Forest University, Winston-Salem, North Carolina, United States of America

* smg632@psu.edu

**Data Availability Statement:** All supporting files are available from the Open Science Foundation database (DOI 10.17605/OSF.IO/MV6EK).

**Funding:** The author(s) received no specific funding for this work.

## Abstract

Having a secure sense of belonging at school supports students' academic achievement and well-being. However, little research has examined how students' personalities relate to their feelings of school belonging. We address this gap in the literature by leveraging data from a large sample of first-year college students (*N* = 4,753) from a diverse set of North American colleges and universities (*N* = 12). We found that both extraversion and agreeableness were positively associated with belonging, while neuroticism was negatively associated with belonging. In an exploratory analysis, we examined differences between large and small schools. Students who were more extraverted, less neurotic, and less open were more likely to attend large schools. Additionally, the association between extraversion and belonging was stronger for students at large schools. These findings advance our understanding of who comes to feel like they belong at college and how school context may influence these relationships. We emphasize the need for continued research on the relationship between personality and belonging. Additionally, we highlight the implications of these results for higher education institutions.

## Introduction

Students' sense of belonging at school matters for their academic performance, school experience, and well-being. Both correlational and experimental evidence link higher school belonging to greater academic achievement, campus engagement, use of support services, cultivation of close friend and mentor relationships, health, and psychological well-being [1–4]. Students' feelings of belonging may be especially important during academic transitions, such as the first year of college, when students are navigating new challenges for the first time [5].

Given the importance of belonging for students' experiences and outcomes, it is valuable to understand personal qualities that are associated with belonging. In the present research, we focus on students' personalities. For example, students high in extraversion (extraverts) tend to be more assertive [6, 7] and successful in forming satisfying interpersonal relationships [8], even suffering when they cannot pursue their inherent urge for social contact [9]. In contrast, students low in extraversion (introverts) experience higher rates of social anxiety [10] and are

**Competing interests:** The authors have declared that no competing interests exist.

less likely to draw upon social support in an effort to cope with their problems [11]. Thus, extraverts may be more likely to act in ways that help them connect with peers, integrate on campus, and develop a secure sense of belonging. This example illustrates how personality could reasonably be associated with how students engage on campus, how they make sense of their college experiences, and whether or not they feel like they belong.

Despite these plausible connections, very little research has examined the association between personality and belonging at any level of schooling. Therefore, the present study examines how students' personality relates to their school belonging during the transition to college, a critical time in students' assessment of their fit with their institution. Further, we consider the possibility that these associations may be shaped by the college context.

## Belonging in college

Walton and Brady [12] define belonging as "[a] general inference, drawn from cues, events, experiences, and relationships, about the quality of fit or potential fit between oneself and a setting. It is experienced as a feeling of being accepted, included, respected. . ." (p. 272). The desire for belonging is a fundamental human need that drives behavior, confers great benefits to those who achieve it, and—when thwarted—is related to an array of ill effects on performance, adjustment, and well-being [13].

In educational settings, a secure sense of belonging confers many benefits to students. For example, college students' sense of belonging at their institution (hereafter, "school belonging" or simply "belonging") is positively associated with their academic self-efficacy, intrinsic motivation, and perceptions of the value of academic assignments [14]. In a nationally representative sample of US college students at four-year institutions, Gopalan and Brady [2] found that students' belonging at the end of their first year of college was positively associated with their persistence, mental health, and use of campus services two years later. Additionally, a secure sense of school belonging reduces students' likelihood to consider dropping out [15].

Evidence from randomized field experiments indicates that benefits of belonging can be causal. Most of these experiments employ the social-belonging intervention [16, 17], a brief reading and writing exercise designed to mitigate students' concerns about belonging by contextualizing everyday difficulties in the transition to college (e.g., feeling lonely, receiving critical feedback on schoolwork) and emphasizing that these challenges are common and typically lessen over time. For example, in a diverse sample of first-year students from a broad-access university, Murphy et al. [18] found that a social-belonging intervention increased minoritized and first-generation college students' likelihood of continuous enrollment over the following two years by nine percentage points. Other studies have documented benefits in a wide variety of contexts on outcomes such as academic performance [4, 19], health and well-being [4], and school integration [for reviews, see 16, 19, 20].

## Personality as a quality that may matter for school belonging

Fundamentally, if belonging is an assessment of one's fit in a particular setting [12], then it is an inference that seeks to answer the question, "Do *I* belong *here*?". Answering this question requires taking stock of one's self ("I"), one's setting ("here"), and the congruence between the two.

To date, the vast majority of the research on the "self" portion of this question has focused on the ways in which a person's sociodemographic group identities like gender, race, and social class shape a person's perception of their fit in a particular space or domain. For example, environmental cues that evoke masculinity (e.g. a Star Trek poster on a wall) can deter women's interest in computer science, in part by making them feel like they won't belong

there [21]. Neutral environmental cues, such as nature posters, don't have this same effect [12]. In a second example, although both White students and students of color anticipate greater belonging in a college major when they believe it is more populated with people from their own racial-ethnic background, this effect is stronger for students of color [22].

However, when people think about who they are, they often focus not only on their socio-demographic identities, but also on their personality [23]. Personality spontaneously comes to mind when people describe themselves, and it undergirds people's narrative sense of their identity [24]. Additionally, personality at one point predicts people's subsequent life goals and life narratives [25]. Consistent with this, Lounsbury et al. [26] found that a third of the variance in sense of identity was accounted for by personality.

Although there are many conceptualizations of personality, this paper will use the Big Five framework [27] which outlines five primary personality dimensions. These include extraversion (To what extent is a person outgoing, sociable, and uninhibited?), neuroticism (To what extent do they get nervous easily or handle stress poorly?), agreeableness (To what extent are they generally trusting, friendly, and accepting of others?), conscientiousness (To what extent are they thorough and dependable?) and openness (To what extent do they have an active imagination and many artistic interests?) [28]. These dimensions are intentionally broad, capturing personality at a high level [27].

In the United States, college is traditionally framed as a place to develop your independent self [29] and thus very relevant to personality. Past research on how students' personalities are associated with their school lives has largely focused on personality as a predictor of academic achievement [e.g., 30]. For example, a recent meta-analysis found conscientiousness to be a strong and robust predictor of academic performance as measured by students' grades, exam performance, GPA, or standardized test performance [31]. Even when controlling for cognitive ability; it accounted for 28% of the explained variance in these outcomes.

But personality can also matter for other educational outcomes, such as time to graduation and even creativity. With regard to time to graduation, students high in conscientiousness tend to graduate in less time than their peers, while students high in agreeableness tend to take longer to graduate [32]. This is theorized to be due to the fact that those who are highly conscientious are more hardworking and persistent and those who score high on agreeableness are less focused on graduation and more on pro-social activities. With regard to creativity, Sung and Choi [33] found that extraversion and openness both positively predict the generation of novel and potentially useful ideas.

There are at least two possible ways in which personality could relate to school belonging. First, it could be that one's personality at the beginning of college affects whether and how they engage in efforts to cultivate belonging felt at the end of their first year. This is illustrated in the example at the beginning of the paper, with regard to extraverts being more likely to act in ways that bolster their sense of belonging. One could imagine an opposite scenario for students high in neuroticism: To the extent that a person gets nervous easily and handles stress poorly, they may find it difficult to cultivate a sense of belonging.

Second, it could be that one's personality at the end of their first year of college influences their belonging felt at the same time point. For example, maybe students with higher agreeableness simultaneously feel a greater sense of belonging. Perhaps highly agreeable individuals are more accepting of others, and see themselves as more similar to others, and thus feel as though they fit in more.

## Existing research on associations between personality and school belonging

Despite burgeoning research on personality development [34–36], evidence of the association between students' personality and their school belonging is limited and solely focused on extraversion. Indeed, this existing research is composed of only three studies across two papers [37, 38]. Both of them find extraversion to be positively associated with school belonging for college students.

**Limitations of existing research.** In addition to being scant, the existing research on personality and belonging in college is limited in a number of ways. First, the existing studies focus only on a single personality dimension [i.e., extraversion; 37, 38] rather than considering all five personality dimensions simultaneously. Given the covariation between personality dimensions [39], we are left with an incomplete picture of how personality relates to belonging.

Second, the studies include relatively few students from only three institutions [37, 38]. The total combined sample size is only 1,112 students. Moreover, the two institutions from which Harris et al. [37] gathered their sample were Harvard and Stanford, which are vastly different from the kinds of institutions the majority of college students attend. This raises the question about whether the relationships observed will generalize to students at other institutions. A further consequence of this limitation is that studies have not yet been able to explore whether —as we allude to above—school context matters for the associations between personality dimensions and belonging.

**Broader research on associations between personality and belonging.** Given the limitations of the research specifically on personality and school belonging, we extrapolated from broader research on personality and other types of belonging—such as "general" belonging [feelings of connectedness, non-specific to the college setting; 40] or "social connectedness" [feelings of fitting in with peers; 41]—to theorize about the likely associations between students' personality and their school belonging.

These broader studies also find a positive relationship between extraversion and belonging [40, 42–47], as well as between extraversion and belonging-relevant activities like social media use [48, 49] and volunteering [50, 51]. Additionally, two other consistent patterns emerge: a negative relationship between neuroticism and belonging [40, 42, 47] and a positive relationship between agreeableness and belonging [40, 42, 43, 46, 47]. The findings regarding how conscientiousness and openness relate to belonging are both more sparse and less consistent. Most available studies find a positive association between conscientiousness and general belonging [40, 42, 46, 47], while Rodger and Johnson [45] find a negative association. With regard to openness, two studies find a positive relationship with general belonging [40, 46], but two other studies find no significant relationship between these variables [42, 47].

While these studies do not specifically focus on school belonging, they may still be informative for the present research. First, many of these studies appear to use convenience samples of college students [40, 42, 44–46], meaning that although their measures are not of "belonging at college" that may be primarily what students are implicitly reporting. Moreover, measures of school belonging, general belonging, and social connectedness all include items about fitting in, feeling a sense of belonging, and (not) feeling like an outsider. For example, Harris et al. [37]'s school belonging measure included the item "I fit in really well at Stanford" while Gebauer et al. [43] measured general belonging with items such as "How much do you feel that you fit in with your peers?".

### The present study

The present study aims to examine the relationship between students' personalities and their sense of school belonging during their first year of college. We use a large and diverse sample of college students ($N = 4,753$) from 12 different four-year colleges in the United States and Canada, ranging from large public research universities to small private liberal arts colleges, to address some of the limitations of existing research. Students completed survey items about their personality at both the beginning and the end of their first year of college and survey items about their feelings of belonging at college at the end of their first year.

**Research questions.** First, we asked: "How do students' personalities the summer before their first year of college relate to their sense of belonging at the end of their first year?" Second, we asked, "How do students' personalities at the end of their first year of college relate to their sense of belonging at that same time?"

In an effort to contribute to emerging discussions about heterogeneity [52] and how school contexts can shape social psychological processes [53, 54], we also explored whether school context—specifically, whether the student attended a larger school or a smaller school—moderated any of the associations between the personality dimensions and belonging.

**Hypotheses.** We hypothesized that extraversion would be positively associated with belonging and neuroticism would be negatively associated with belonging. We did not make specific predictions for agreeableness, conscientiousness, or openness.

**Contributions.** At a theoretical level, this research increases our knowledge of the relationship between students' personality and their feelings of school belonging. Furthermore, it sheds light on how school context may affect these relationships. At a practical level, these findings could inform efforts to increase feelings of belonging in all students, thus offering each student a better chance at success in college.

## Method

### Data source and procedure

This research uses data from the College Transition Collaborative (CTC) Social-Belonging Multi-Site Randomized Control Trial, a large study of students who entered a diverse set of postsecondary institutions in 2015 and 2016 [55]. The primary purpose of the overarching study was to test social-belonging interventions [4, 16] across diverse post-secondary contexts. However, it also provided researchers an opportunity to collect brief measures related to other aspects of students' experience in the transition to college, as was the case for the personality measures collected for the present study.

The current study involved only the analysis of de-identified secondary data. Therefore, Institutional Review Board (IRB) approval was deemed unnecessary. However, the overarching study was approved by the IRB at Stanford University and either approved by the IRB at each of the colleges from which data were collected or deemed by them to not need review (see Walton et al. [53] for more details). For the purpose of this study, the data was first accessed on July 28th, 2021. Before conducting analyses, we preregistered our study design, inclusion/exclusion criteria, and planned analyses. The preregistration is available on OSF (https://osf.io/mv6ek/), along with materials, data, and code. Any deviations from the preregistration are reported in the text. We report all manipulations, measures, and exclusions in the present study.

The summer before their first year of college (Time 1; $T_1$), students were invited by their institution (e.g., via orientation checklist, email invitations, etc.) to participate in an online "activity" on students' experiences in the transition to college. Embedded in the activity was a social belonging intervention. The details of the intervention are not relevant to the present

paper, except that we control for which version of the intervention a student completed: control, standard treatment, or locally customized treatment. [For more information on intervention treatment effects, see 53, 56, 57]. After completing their assigned version, students were asked to answer demographic and individual difference measures, some of which were the personality measures for the present study. There was no compensation for participation.

At the end of their first year of college (Time 2; $T_2$), a subsample of students who participated during the prior summer were invited by their institution to participate in a survey on their social and academic experiences during the past school year. The personality and belonging measures were embedded in the survey. Some students received compensation for completing this survey; the presence, amount, and type of compensation varied by institution.

## Participants

The study focused on students who (a) were a first-year (non-transfer) student, (b) completed the personality scale at least once, and (c) completed the belonging scale in the survey at the end of their first year. In total, 4,753 students from 12 schools met these criteria. Not all students answered all of the questions relevant for each research question, most commonly because the relevant measures were not assessed at their school at the time point in question. Therefore, the number of students and schools included in analyses differs across the two research questions.

See Tables 1 and 2 for information about the participating schools as well as for information regarding the participant sample at each school. Overall, students from nine schools are included in the analyses for the first research question ($N$ = 2,137), students from 11 schools are included in the analyses for the second research question ($N$ = 4,262), and students from seven schools are included in the analyses for the third research question ($N$ = 1,646).

## Measures

**Personality.** The Big Five Inventory-10 [BFI-10; 28] was used to assess personality at both $T_1$ and $T_2$. In the BFI-10, each personality dimension is assessed with two items. Students rate the items on a 5-point scale ranging from 1 (*strongly disagree*) to 5 (*agree strongly*). This measure was selected because of its validity and efficiency [58], knowing that the gain in efficiency often results in lower reliability (see Duckworth and Yeager [59] for a discussion of practical measurement). Cronbach alpha reliabilities (extraversion: $T_1$ = .69, $T_2$ = .70; neuroticism: $T_1$ = .57, $T_2$ = .56; agreeableness: $T_1$ = .32, $T_2$ = .36; conscientiousness: $T_1$ = .48, $T_2$ = .49; openness: $T_1$ = .22, $T_2$ = .34) were in line with previous research on the BFI-10 [28, 60–62]. The test-retest reliabilities for each dimension (intraclass correlation coefficients: extraversion = .69, neuroticism = .63, agreeableness = .53, conscientiousness = .59, openness = .53) were considered moderate [63].

**Belonging.** At $T_2$, students answered four items about their sense of belonging at their school (adapted from Walton and Cohen [17]; sample item: "I feel I belong at [school name]"; scale: 1 = *strongly disagree* to 6 = *strongly agree)*. The measure was highly reliable ($\alpha$ = .89).

**Covariates.** In our analyses, we included covariates for the following variables: school attended, intervention condition, race, gender, and college generation status. Our preregistration did not include generation status as a covariate. However, we decided to include this characteristic to control for the effects of being a first- or continuing-generation student (i.e., first-generation: no parents/guardians had earned a 4-year college degree; continuing-generation: at least one parent/guardians had earned a 4-year college degree).

In each case, we used the largest category as the reference group: as such, the intercept represents continuing-generation White women in the control condition who were enrolled at

**Table 1. Institutional characteristics by school.**

| School | Sector | Type | Location | Undergrad enrollment | N in Analytic Sample | Included in RQ1 Analysis | Included in RQ2 Analysis |
|--------|--------|------|----------|---------------------|---------------------|--------------------------|--------------------------|
| A | Private | Liberal Arts | Midwest US | 1,000–4,999 | 134 | - | Y |
| B | Private | Liberal Arts | Northeast US | 1,000–4,999 | 274 | Y | Y |
| C | Private | Liberal Arts | Midwest US | 1,000–4,999 | 470 | Y | Y |
| D | Private | Liberal Arts | Midwest US | 1,000–4,999 | 352 | Y | Y |
| E | Public | Research | Midwest US | >5,000 | 604 | Y | Y |
| F | Public | Research | Midwest US | >5,000 | 953 | Y | Y |
| G | Private | Liberal Arts | Midwest US | 1,000–4,999 | 187 | - | Y |
| H | Private | Liberal Arts | Northwest US | 1,000–4,999 | 463 | Y | Y |
| I | Private | Liberal Arts | Midwest US | 1,000–4,999 | 167 | - | Y |
| J | Private | Liberal Arts | Midwest US | <1,000 | 197 | Y | Y |
| K | Public | Research | Canada | >5,000 | 442 | Y | - |
| L | Private | Liberal Arts | Midwest US | 1,000–4,999 | 510 | Y | Y |
| *Total* | - | - | - | - | 4,753 | 9 | 11 |

the largest school in the sample, a public state university. Overall, we included up to eleven dummy variables for school, two for condition assignment (standard, customized), seven for race (Native, Black, Hispanic, Asian, other, multiracial, unspecified), two for gender (man, other) and two for generation status (first generation, unknown). The codes were mutually exclusive for all covariates.

**Analytic plan.** To examine whether personality at the summer before college ($T_1$) and personality at the end of the first year of college ($T_2$) were associated with belonging at the end of the first year of college ($T_2$), we used multivariate regression. For each research question, we first ran a personality-only model to examine the unadjusted relationships between the personality variables and belonging. Then, in our preferred model, we included both the personality variables and the covariates specified above as predictors. Finally, we ran a covariate-only model without any personality variables and compared this model with our preferred model

**Table 2. Student participant demographics by school.**

| School | Mean Age[a] | Gender | | | First-Gen (%) | Racial-Ethnic Identity | | | | | | | |
|--------|------|-----------|-------------|-----------|---------------|-----------|-----------|-----------|--------------|------------|-----------|-----------|------------------|
| | | Man (%) | Woman (%) | Other (%) | | White (%) | Black (%) | Asian (%) | Hispanic (%) | Native (%) | Multi (%) | Other (%) | Undisclosed (%) |
| A | 18.23 | 26.12 | 71.64 | 2.24 | 32.84 | 63.43 | 9.70 | 6.72 | 8.96 | 0.00 | 6.72 | 4.48 | 0.00 |
| B | 18.38 | 29.56 | 66.42 | 4.01 | 25.55 | 72.99 | 7.66 | 5.11 | 6.20 | 0.36 | 4.38 | 3.28 | 0.00 |
| C | 18.46 | 40 | 58.51 | 1.49 | 27.45 | 67.45 | 5.74 | 13.40 | 6.81 | 0.85 | 2.34 | 3.40 | 0.00 |
| D | 18.39 | 25.85 | 74.15 | 0.00 | 19.89 | 81.53 | 3.69 | 3.41 | 5.97 | 0.00 | 1.70 | 3.69 | 0.00 |
| E | 18.53 | 20.36 | 77.32 | 2.32 | 45.03 | 60.43 | 24.01 | 0.83 | 2.81 | 0.50 | 2.32 | 9.11 | 0.00 |
| F | 18.45 | 30.33 | 68.42 | 1.25 | 18.68 | 74.92 | 6.09 | 6.30 | 5.14 | 0.73 | 2.31 | 4.30 | 0.21 |
| G | 18.28 | 33.69 | 63.10 | 3.20 | 21.39 | 61.50 | 9.09 | 11.76 | 10.16 | 0.53 | 3.74 | 2.14 | 1.07 |
| H | 18.40 | 24.84 | 73.00 | 2.16 | 15.33 | 71.06 | 2.59 | 7.13 | 8.64 | 2.38 | 4.32 | 3.67 | 0.22 |
| I | - | 35.93 | 59.88 | 4.19 | 31.14 | 70.66 | 7.78 | 4.79 | 6.59 | 0.60 | 6.59 | 2.40 | 0.60 |
| J | 18.66 | 96.45 | 0.00 | 3.55 | 39.59 | 72.08 | 7.11 | 8.63 | 5.08 | 0.00 | 1.52 | 4.57 | 1.02 |
| K | - | 38.24 | 59.28 | 2.49 | 19.91 | 42.53 | 2.26 | 50.68 | 1.13 | 0.00 | 0.00 | 2.71 | 0.68 |
| L | - | 37.06 | 61.57 | 1.37 | 15.10 | 66.67 | 9.22 | 11.57 | 3.73 | 0.20 | 3.14 | 4.90 | 0.59 |
| *Overall* | 18.44 | 33.52 | 64.49 | 1.99 | 24.59 | 67.33 | 8.21 | 11.07 | 5.30 | 0.61 | 2.76 | 4.44 | 0.29 |

[a] Three schools did not collect data used to compute age.

using an ANOVA to understand the extent to which our preferred model explained variance above and beyond student and school characteristics.

Our preregistration included two additional research questions, one about personality change and another about the relationship between personality change and belonging. Additionally, our preregistration specified two sets of exploratory analyses, one mirroring the primary analyses but with belonging uncertainty (instead of belonging) as the outcome and another considering whether intervention condition moderates the relationship between personality and belonging. Results for these additional research questions are reported in the Supplemental Materials [S1 File].

Primary and exploratory analyses were performed in R [64] using packages dplyr [65], psych [66], and Matrix [67]. Scale reliabilities were computed in R [64] using package psych [66].

## Results

### Preliminary analyses

Table 3 reports descriptive statistics and zero-order correlations for the key variables. Preliminary analyses were performed in R [64] using package Hmisc [68].

### Association between students' personality at the beginning of college and their belonging at the end of first year

In the personality-only model, extraversion, agreeableness, and conscientiousness at the beginning of students' first year were associated with their belonging at the end of their first year (Table 4). Specifically, $T_1$ extraversion, $T_1$ agreeableness, and $T_1$ conscientiousness ($b = .085$, $se = .027$, $t = 3.201$, $p = .001$) were each positively associated with $T_2$ belonging; the higher a student's extraversion, agreeableness, or conscientiousness during the summer before college, the higher their sense of belonging at the end of their first year of college. In addition, there was a marginal effect such that $T_1$ neuroticism was negatively associated with $T_2$ belonging ($b = -.041$, $se = .021$, $t = -1.923$, $p = .055$); the lower a student's neuroticism the summer before college, the higher their sense of belonging at the end of their first year of college. This model explained 5.2% of the variance in $T_2$ belonging ($R^2 = .052$, $F(5, 2131) = 24.560$, $p < .001$).

In our preferred model which controlled for school and personal characteristics, the positive associations discussed above for $T_1$ extraversion and $T_1$ agreeableness held (Table 4). The positive association between $T_1$ conscientiousness and $T_2$ belonging was smaller in magnitude and of marginal statistical significance ($b = .045$, $se = .027$, $t = 1.671$, $p = .095$). In addition, the negative association between $T_1$ neuroticism and $T_2$ belonging increased in statistical significance ($b = -.060$, $se = .022$, $t = -2.722$, $p = .007$). This model explained 8.8% of the variance in $T_2$ belonging ($R^2 = .088$, $F(26, 2110) = 8.967$, $p < .001$). In addition, students who identified racial-ethnically as Black, Asian, or "Other" reported lower $T_2$ belonging compared to White students.

Our preferred model that included the covariates in addition to personality had greater predictive validity than the covariate-only model ($F(5, 2115) = 22.827$, $p < .001$). Including personality in the model more than doubled the percent variance explained by the covariates alone, explaining an additional 4.7% of the variance in students' $T_2$ belonging.

### Association between students' personality and belonging at the end of first year

In the personality-only model, extraversion, agreeableness, conscientiousness, and neuroticism at the end of students' first year were associated with their belonging at the end of their

**Table 3. Descriptive statistics and zero-order correlations.**

|  |  | Descriptive Statistics | | | Zero-Order Correlations | | | | | | | | | | |
|---|---|---|---|---|---|---|---|---|---|---|---|---|---|---|---|
|  |  | N | Mean | SD | 1 | 2 | 3 | 4 | 5 | 6 | 7 | 8 | 9 | 10 | 11 |
| **Summer Before College (T1)** | | | | | | | | | | | | | | | |
| 1 | Extraversion | 2,138 | 3.16 | 1.06 | - | -0.31*** | 0.23*** | 0.15*** | 0.03 | 0.70*** | -0.22*** | 0.21*** | 0.09*** | 0.01 | 0.18*** |
| 2 | Neuroticism | 2,137 | 3.08 | 1.01 | - | - | -0.12*** | -0.15*** | 0.01 | -0.24*** | 0.64*** | -0.10*** | -0.07** | 0.03 | -0.11*** |
| 3 | Agreeableness | 2,137 | 3.81 | 0.81 | - | - | - | 0.20*** | 0.02 | 0.17*** | -0.08** | 0.54*** | 0.13*** | 0.00 | 0.16*** |
| 4 | Conscientiousness | 2,137 | 3.76 | 0.79 | - | - | - | - | 0.04 | 0.09*** | -0.08** | 0.15*** | 0.60*** | -0.02 | 0.12*** |
| 5 | Openness | 2,138 | 3.58 | 0.91 | - | - | - | - | - | -0.02 | 0.08** | -0.03 | 0.01 | 0.53*** | 0.00 |
| **End of First Year (T2)** | | | | | | | | | | | | | | | |
| 6 | Extraversion | 4,262 | 3.13 | 1.08 | - | - | - | - | - | - | -0.28*** | 0.20*** | 0.13*** | 0.03 | 0.24*** |
| 7 | Neuroticism | 4,262 | 3.18 | 1.02 | - | - | - | - | - | - | - | -0.11*** | -0.11*** | 0.06*** | -0.14*** |
| 8 | Agreeableness | 4,263 | 3.71 | 0.87 | - | - | - | - | - | - | - | - | 0.13*** | 0.01 | 0.20*** |
| 9 | Conscientiousness | 4,262 | 3.65 | 0.82 | - | - | - | - | - | - | - | - | - | 0.00 | 0.12*** |
| 10 | Openness | 4,262 | 3.61 | 0.93 | - | - | - | - | - | - | - | - | - | - | 0.02 |
| 11 | Belonging | 4,753 | 4.73 | 0.98 | - | - | - | - | - | - | - | - | - | - | - |

The personality dimensions are measured on 5-point scales. Belonging is measured on a 6-point scale.

$^{*}p < .05.$

$^{**}p < .01.$

$^{***}p < .001.$

**Table 4. Regression table: $T_1$ Personality Predicting $T_2$ Belonging.**

|  | Personality-only | | Preferred | | Covariate-only | |
|---|---|---|---|---|---|---|
|  | B (SE) | 95% CI | B (SE) | 95% CI | B (SE) | 95% CI |
| **Extraversion** | **.119*** (.021)** | **[.078, .159]** | **.104*** (.021)** | **[.064, .145]** | - | - |
| Neuroticism | -.041 (.021) | [-.083, .001] | -.060** (.022) | [-.104, -.017] | - | - |
| Agreeableness | 0.133*** (0.026) | [.082, .185] | .153*** (.026) | [.102, .204] | - | - |
| Conscientiousness | .085** (.027) | [.033, .138] | .045 (.027) | [-.008, .098] | - | - |
| Openness | -.005 (.022) | [-.049, .039] | .009 (.023) | [-.035, .054] | - | - |
| Black | - | - | -.585*** (.082) | [-.746, -.424] | -.519*** (.083) | [-.682, -.356] |
| Asian | - | - | -.138* (.064) | [-.263, -.013] | -.183** (.064) | [-.309, -.057] |
| Hispanic | - | - | -.159 (.097) | [-.350, .032] | -.133 (.099) | [-.328, .061] |
| Native | - | - | -.418 (.280) | [-.968, .131] | -.393 (.287) | [-.956, .170] |
| Multiracial | - | - | -.957 (.928) | [-2.776, .862] | -.678 (.948) | [-2.538, 1.182] |
| Other Race | - | - | -.211* (.090) | [-.387, -.035] | -.226* (.092) | [-.406, -.046] |
| Unspecified Race | - | - | .329 (.416) | [-.486, 1.144] | .230 (.426) | [-.604, 1.065] |
| Man | - | - | -.038 (.046) | [-.129, .053] | -.023 (.046) | [-.113, .067] |
| Other Gender | - | - | -.082 (.156) | [-.389, .224] | -.197 (.159) | [-.509, .115] |
| First-Generation Student | - | - | -.022 (.049) | [-.117, .074] | -.021 (.050) | [-.119, .077] |
| Unknown Generation Status | - | - | .026 (.242) | [-.449, .501] | -.001 (.247) | [-.487, .484] |
| Adjusted $R^2$ | 0.052 | | 0.088 | | 0.041 | |
| N Observations | 2,137 | | 2,137 | | 2,137 | |

The reference category is White, continuing-generation women in the control condition. To economize on space, we do not report the dummy variables for each school and condition (available from authors upon request).

$^{*}p < .05.$

$^{**}p < .01.$

$^{***}p < .001.$

first year (Table 5). Specifically, $T_2$ extraversion, $T_2$ agreeableness, and $T_2$ conscientiousness were each positively associated with $T_2$ belonging; the higher a student's extraversion, agreeableness, or conscientiousness at the end of their first year of college, the higher their sense of belonging at that same time. In contrast, $T_2$ neuroticism was negatively associated with $T_2$ belonging; the lower a student's neuroticism at the end of their first year of college, the higher their sense of belonging at that same time. This model explained 9.2% of the variance in $T_2$ belonging ($R^2 = .092$, $F(5, 4256) = 86.850$, $p < .001$).

In our preferred model which controlled for school and personal characteristics, the positive associations discussed above for $T_2$ extraversion, $T_2$ agreeableness, and $T_2$ conscientiousness held, as did the negative association for $T_2$ neuroticism (Table 5). This model explained 13% of the variance in $T_2$ belonging ($R^2 = .130$, $F(28, 4233) = 23.770$, $p < .001$). In addition, Black, Asian, Hispanic, Native, Multiracial and first-generation students reported lower $T_2$ belonging compared to their White and continuing-generation peers.

Our preferred model that included the covariates in addition to personality had greater predictive validity than the covariate-only model ($F(5, 4238) = 84.656$, $p < .001$). Including personality in the model nearly tripled the percent variance explained by the covariates alone, explaining an additional 8.6% of the variance in students' $T_2$ belonging.

### Exploratory analyses: Moderation by school type

For our exploratory analyses regarding moderation by school type, we grouped the schools into two categories: (1) large, research-intensive 4-year colleges and (2) smaller, liberal arts colleges. Before undertaking the main moderation analyses, we first examined whether students differed in their personalities between the large and small schools (Table 6). We find that, at $T_1$, students enrolled in large schools in our sample are significantly more extraverted, less neurotic, and less open. Interestingly, we do not observe statistically significant differences between students' personality dimensions at large vs. small schools at the end of their first year ($T_2$).

We then interacted school type ("large school" *Yes = 1; No = 0*) with the personality dimensions across both time points to predict students' belonging at the end of their first year. All other model specifications (including the use of student- and school-level covariates) are similar to the primary analyses. These exploratory analyses were conducted in Stata/SE Version 15.1 [69].

A few notable results emerge (Table 7). First, we find that across both time points, the magnitude and direction of the associations between personality and belonging are fairly similar across the two school types. Specifically, $T_1$ extraversion ($b = .115$, *se* $= .029$, $t = 4.03$, $p < .001$), $T_2$ extraversion ($b = .142$, *se* $= .018$, $t = 8.12$, $p < .001$), $T_1$ agreeableness ($b = .179$, *se* $= .036$, $t = 4.91$, $p < .001$), and $T_2$ agreeableness ($b = .171$, *se* $= .021$, $t = 8.17$, $p < .001$) were each positively associated with $T_2$ belonging. However, associations between (a) $T_1$ openness, (b) $T_2$ openness, (c) $T_1$ conscientiousness, (d) $T_2$ conscientiousness, (e) $T_1$ neuroticism and $T_2$ belonging are less precise.

Second, we find statistically significant moderations by school type when it comes to extraversion. The association between extraversion and belonging, especially at $T_2$, is stronger for students in large schools ($T_1$ Extraversion x Large Schools $b = .030$, *se* $= .042$, $t = -.72$, $p = .475$; $T_2$ Extraversion x Large Schools $b = .062$, *se* $= .029$, $t = 2.13$, $p = .033$). Being more extraverted might help students in larger, research-intensive colleges cultivate a stronger sense of school belonging.

**Table 5. Regression table: $T_2$ Personality Predicting $T_2$ Belonging.**

| | Personality-only | | Preferred | | Covariate-only | |
|---|---|---|---|---|---|---|
| | *B (SE)* | *95% CI* | *B (SE)* | *95% CI* | *B (SE)* | *95% CI* |
| **Extraversion** | **.169\*\*\* (.014)** | **[.141, .197]** | **.165\*\*\* (.014)** | **[.064, .145]** | - | - |
| **Neuroticism** | -.066\*\*\* (.015) | [-.095, -.037] | -.082\*\*\* (.015) | [-.104, -.017] | - | - |
| **Agreeableness** | .165\*\*\* (.017) | [.131, .198] | .163\*\*\* (.017) | [.102, .204] | - | - |
| **Conscientiousness** | .085\*\*\* (.018) | [.050, .119] | .047\*\* (.018) | [-.008, .098] | - | - |
| **Openness** | .018 (.016) | [-.013, .048] | .023 (.015) | [-.035, .054] | - | - |
| **Black** | - | - | -.449\*\*\* (.053) | [-.746, -.424] | -.428\*\*\* (.055) | [-.536, -.320] |
| **Asian** | - | - | -.215\*\*\* (.057) | [-.263, -.013] | -.255\*\*\* (.059) | [-.371, -.138] |
| **Hispanic** | - | - | -.236\*\*\* (.063) | [-.350, .032] | -.232\*\*\* (.066) | [-.362, -.102] |
| **Native** | - | - | -.394\* (.172) | [-.968, .131] | -.475\*\* (.180) | [-.829, -.121] |
| **Multiracial** | - | - | -.257\*\* (.083) | [-2.776, .862] | -.271\*\* (.087) | [-.441, -.101] |
| **Other Race** | - | - | -.101 (.069) | [-.387, -.035] | -.064 (.072) | [-.205, .077] |
| **Unspecified Race** | - | - | .060 (.279) | [-.486, 1.144] | -.123 (.292) | [-.696, .450] |
| **Man** | - | - | -.040 (.034) | [-.129, .053] | -.034 (.033) | [-.099, .032] |
| **Other Gender** | - | - | -.064 (.105) | [-.389, .224] | -.231\* (.109) | [-.446, -.017] |
| **First-Generation Student** | - | - | -.089\* (.036) | [-.117, .074] | -.094\* (.037) | [-.167, -.021] |
| **Unknown Generation Status** | - | - | -.106 (.064) | [-.449, .501] | -.125 (.067) | [-.258, .007] |
| **Adjusted $R^2$** | 0.092 | | 0.130 | | 0.044 | |
| **N Observations** | 4,262 | | 4,262 | | 4,262 | |

The reference category is White, continuing-generation women in the control condition. To economize on space, we do not report the dummy variables for each school and condition (available from authors upon request).

\*p < .05.

\*\*p < .01.

\*\*\*p < .001.

## Discussion

In a large, multi-institution sample with longitudinal data we find that college students' personalities do relate to their school belonging at the end of their first year. Specifically, students' belonging is consistently associated with the extent to which they are extraverted, agreeable,

**Table 6. Personality: Large schools vs. small schools.**

| | Large Schools Predicting T1 Personality | Large Schools Predicting T2 Personality |
|---|---|---|
| **Extraversion** | 0.267\*\* (0.101) | -0.029 (0.100) |
| **Neuroticism** | -0.190\* (0.094) | -0.144 (0.091) |
| **Agreeableness** | -0.047 (0.078) | -0.0760 (0.080) |
| **Conscientiousness** | 0.060 (0.075) | 0.096 (0.075) |
| **Openness** | -0.244\*\* (0.087) | -0.039 (0.086) |
| **N Observations** | 2,137 | 4,262 |

Standard errors in parentheses. Each cell reports the "Large Schools" coefficient from separate, covariate-adjusted, regressions each predicting the personality dimensions shown in row A. The reference category is White, continuing-generation women in the control condition. To economize on space, we report only the key coefficients.

\*p < .05.

\*\*p < .01.

\*\*\*p < .001.

**Table 7. Moderations by school type.**

| | T1 Personality Predicting T2 Belonging | T2 Personality Predicting T2 Belonging |
|---|---|---|
| **Extraversion** | .115*** (.029) | .142*** (.018) |
| **Neuroticism** | -.046 (.030) | -.101*** (.019) |
| **Agreeableness** | .179*** (.036) | .171*** (.021) |
| **Conscientiousness** | .006 (.037) | .038 (.022) |
| **Openness** | .005 (.032) | .015 (.019) |
| **Large Schools** | .187 (.368) | -.437 (.062) |
| **Extraversion x Large Schools** | -.030 (.042) | .062* (.029) |
| **Neuroticism x Large Schools** | -.034 (.043) | .052 (.030) |
| **Agreeableness x Large Schools** | -.057 (0.052) | -.022 (0.035) |
| **Conscientiousness x Large Schools** | .084 (.053) | .034 (.037) |
| **Openness x Large Schools** | .003 (.045) | .025 (.032) |
| **_N_ Observations** | 2,137 | 4,262 |

Standard errors in parentheses. The reference category is White, continuing-generation women in the control condition. To economize on space, we do not report the dummy variables for each school. Each column reports the results from separate, covariate-adjusted, regressions where "Large Schools" indicator is interacted with each of the personality dimensions to predict T2 Belonging.

*$p < .05$.

**$p < .01$.

***$p < .001$.

neurotic, and also—in some cases—the extent to which they are conscientious. Extraversion and agreeableness are positively associated with belonging, while neuroticism is negatively associated with belonging. This is true both longitudinally and concurrently. Conscientiousness is positively associated with school belonging only when they are measured concurrently.

Of note, openness was statistically unrelated to students' school belonging. The overall point estimate of this relationship was very close to 0. On the one hand, this aligns with past research which does not consistently find a relationship between openness and general belonging [40, 42]. On the other hand, given the wide variety of new experiences offered at college, it seems reasonable that students high in openness would take more advantage of these experiences and, thereby, cultivate their sense of belonging. But we don't see evidence of that. One possibility is that students high in openness take advantage of many different opportunities without deeply committing to a few that might better foster their feelings of belonging. Future research should examine these possibilities.

A key contribution of our research is to begin to consider how school characteristics may matter for the relationship between personality and school belonging. Specifically, we find that the associations between extraversion and belonging might be stronger for students at large public research colleges compared with students at smaller liberal arts colleges. Although we were only able to examine one school characteristic—type/size—our efforts align with recent theoretical and empirical work calling for greater attention to effect heterogeneity [52]. Our exploratory analyses suggested that the direction and magnitude of relationships, especially, between extraversion, agreeableness, and belonging was generally consistent across schools. However, stronger associations between extraversion and belonging for students in larger schools at the end of the first year highlights the need to pay close attention to context heterogeneity in institutional efforts to promote students' sense of belonging.

## Strengths, limitations, and future directions

In comparison to past literature examining personality and belonging in college, the present study benefited from a large sample of college students ($N$ = 4,753), 7.5-fold larger than most other studies, two assessments of all five personality dimensions—as opposed to one assessment—spanning the first year of college, and the focus of school belonging rather than general belonging or social connectedness. The sample was also drawn from multiple schools of different types in both the US and Canada, increasing confidence in the generalizability of our findings. While our study addresses most of the limitations of previous studies, it leaves some limitations untouched, indicating next steps for future investigations.

Although the present research is among very few studies that measure personality and school belonging longitudinally, the results here underscore the value of this approach. Future research might profitably measure both personality and belonging over a longer time frame within college—ideally students' entire college careers—with more robust measures of personality and belonging. This would allow for greater consideration of the potentially reciprocal influences between these two aspects of students' selves and the dynamic college experiences that help students make meaning of their fit with their colleges. Additionally, researchers might also consider measuring these characteristics even earlier in students' school trajectory, perhaps in the transition to high school as done by Wentzel et al. [70]. This would allow future research to capture a more complete understanding of students' felt belonging across their education.

A key limitation of the present study is the use of a very brief measure of personality. A first consequence of using such a brief measure of personality is that we are unable to explore the relationship between particular facets (subtraits) of the personality dimensions and belonging [71]. As such, we don't know if (or which) particular facets are driving the associations observed. Such explorations in other domains have been fruitful; for example, the positive association between extraversion and well-being appears primarily driven by activity level and not by other aspects of extraversion, such as gregariousness or warmth [72]. A more fine-grained analysis of this nature for belonging might offer both theoretical and practical insights. A second consequence of using such a brief measure of personality is that the reliabilities for the dimensions were not high (extraversion being the modest exception to this). The brief measure was necessary given the constraints of the larger study in which these data were collected. However, more efficient measures tend to have lower reliability. Reassuringly, however, the literature has suggested that short measures might have advantages in terms of validity, possibly by reducing participants' boredom and fatigue [73]. For example, the predictive validity of the BFI-10 is almost as high as, and sometimes even higher than, that of longer Big Five inventories [58, 59]. Still, future research should use more robust measures of personality for a more nuanced and precise examination of these relationships.

Another limitation in our study is in the depth of the exploratory analyses. Our moderation analysis, while instructive, only focuses on a single school characteristic, size, across 12 different schools. Although this still advances the literature, it is very much an initial examination of how context and personality may interact to affect belonging. Future investigations might explore how different college factors, such as cost and campus life, interact with personality to affect students' sense of belonging at a wider range of institutions.

The present research points to the value of considering the relationship between personality and belonging for students making the transition to college, answering a call to action for better integration of personality and social psychology [74]. It highlights the value of future investigations that could consider addressing how different personalities might lead students to take (or not to take) different actions and how those actions might then contribute to the extent

that students feel a sense of belonging. For example, future research could focus on whether highly agreeable students, who are generally more friendly, might make more friends in their classes and thus experience higher school belonging.

Our study also draws attention to the need for research on the types of actions that institutions could take to better promote sense of belonging in students with certain personality profiles. For example, given the observed association between extraversion and belonging, it's reasonable to wonder if typical institutional interventions to foster belonging tend to be most attractive or supportive for students who are more extraverted.

Furthermore, additional efforts should consider interactions between personality and identity and their effect on belonging, with special attention to students from marginalized backgrounds. For example, how does race/ethnicity or first-generation college status affect the relationship between students' personalities and their sense of belonging at their institution? Perhaps the magnitude of the associations between personality and belonging are blunted for students from negatively stereotyped backgrounds, at least when they attend predominantly White institutions. This could be due to the fact that people, especially White people, find interracial contact stressful [75]. As such, White students—or White instructors—may respond less warmly to an extraverted Black student than they would to an extraverted White student. The less-warm reception could, understandably, interfere with the Black student's development of a secure sense of belonging.

## Implications for practice

Put simply, it may be worthwhile for institutions to consider students' personalities when thinking about efforts to foster belonging. A first step in this direction would be to measure students' belonging and personality to evaluate whether existing efforts are particularly effective for students with a certain personality profile [cf. 76]. For example, it may be that the *ecological belonging intervention* [77] which relies on peer discussions, tends to be more effective for extraverts, because it "fits" with their desire for social connection with others. Alternatively, perhaps this ecological belonging intervention is especially useful for introverts, as they might be less likely to otherwise make these connections with their peers.

A second step would be to consider personality when designing new programming. For example, what might it look like to intentionally cultivate the belonging of students who are higher in neuroticism or lower in agreeableness? While considering questions such as these, it will be important for practitioners to remember that personality is not a fixed trait, but one that is malleable. For example, one study instructing students to act extraverted found results supporting a strong connection between acting extraverted and positive affect [78]. These results suggest the possibility that new college experiences or interventions could alter students' personality, thus providing a pathway for them to develop a stronger sense of belonging.

## Conclusions

As a field, we have come a long way from the polarizing, person-situation debate [74] to a more nuanced understanding of the role of individual and contextual differences combined. Here, we contribute to that understanding by highlighting how personality is related to students' belonging in the first year of college and, further, how those relationships may depend on institution type/size. Ultimately, practitioners should keep in mind that different students—with different backgrounds but also different personalities—may find different avenues to build their sense of belonging on campus. We encourage institutions to create multiple, varied pathways to belonging, and to emphasize that developing a sense of belonging often takes time.

## Supporting information

**S1 File. Supplemental materials.**
(DOCX)

## Acknowledgments

This paper uses data from a larger dataset collected by the College Transition Collaborative (CTC) focused on understanding college students' experiences in the transition to college. This research was made possible through methods and data systems created by the Project for Education Research That Scales (PERTS). We thank Cassie Hartzog and Eranda Jayawickreme for assistance and the full team of CTC researchers, liaisons, and partner schools for making this research possible.

Also, we would like to express our sincere gratitude to the Editor and the two anonymous reviewers for their insightful comments, which have improved the quality of this article.

## Author Contributions

**Conceptualization:** Alexandria M. Stubblebine, Maithreyi Gopalan, Shannon T. Brady.

**Formal analysis:** Alexandria M. Stubblebine, Maithreyi Gopalan, Shannon T. Brady.

**Investigation:** Maithreyi Gopalan, Shannon T. Brady.

**Supervision:** Shannon T. Brady.

**Writing – original draft:** Alexandria M. Stubblebine.

**Writing – review & editing:** Maithreyi Gopalan, Shannon T. Brady.

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
