## [Decision Letter · Decision Letter 0]

4 Oct 2023

PONE-D-23-21304Who Feels Like They Belong? Personality and Belonging in CollegePLOS ONE

Dear Dr. Gopalan,

Thank you for submitting your manuscript to PLOS ONE. After careful consideration, we feel that it has merit but does not fully meet PLOS ONE’s publication criteria as it currently stands. Therefore, we invite you to submit a revised version of the manuscript that addresses the points raised during the review process.

We look forward to receiving your revised manuscript.

Kind regards,

Sohaib Mustafa

Academic Editor

PLOS ONE

Journal Requirements:

2. Please include your tables as part of your main manuscript and remove the individual files. Please note that supplementary tables (should remain/ be uploaded) as separate ""supporting information"" files.

Reviewers' comments:

Reviewer's Responses to Questions

**Comments to the Author**

1. Is the manuscript technically sound, and do the data support the conclusions?

Reviewer #1: Partly

Reviewer #2: Yes

2. Has the statistical analysis been performed appropriately and rigorously? 

Reviewer #1: No

Reviewer #2: Yes

3. Have the authors made all data underlying the findings in their manuscript fully available?

Reviewer #1: Yes

Reviewer #2: Yes

4. Is the manuscript presented in an intelligible fashion and written in standard English?

Reviewer #1: Yes

Reviewer #2: Yes

5. Review Comments to the Author

Reviewer #1: Review PONE-D-23-21304

“Who feels like they belong? Personality and belonging in college”

Although this was a well-written manuscript, my main concern was the examination of the so-called "personality change” (p. 10). Because of the different test-taking contexts, I would argue that the responses to the personality scale cannot be compared between the two time periods. Pages 11-12 explain that students first completed the personality scale before attending university/college. The second session was, “At the end of their first year” (p. 12). The students will be in completely different mind sets for each test-taking time. Time 1 would most likely reflect personality plus eustress or excitement/concern about starting school. Time 2 is probably reflecting personality plus distress because the end of term is either during or possibly at the start of final exams. These differences in context will clearly influence how individuals will respond to the personality items. For example, if anxious about exams, individuals may endorse more neuroticism items and fewer agreeableness items. I would therefore suggest that all mention of “personality change” be removed from the manuscript, especially the analyses and text starting at the bottom of page 16 and Table 5.

Another concern that I have is with Table 6. The text on page 18 states that Table 6 is the mean differences but Table 6 lists regression weights (not the right values).

Additional comments:

Page 9, line 181, I believe it should be “with general” and not “to”.

Page 13, line 276, the alpha symbol is missing (might be easier to simply use the word, “alpha”).

Results section – it is a bit awkward to simply have, for example, “See Table 3.” in the text. Simply put into brackets and connect to the preceding sentence.

Reviewer #2: The differential impact of students' personalities on their sense of belonging is a relevant extension of existing research on students' sense of belonging. The authors had access to a really large number of students to answer their research questions. A closer look at the manuscript revealed that the paper has several strengths, but also some issues that the authors should address and discuss prior to publication. I appreciate the authors' efforts to make their data and scripts openly available. However, I wondered why they were copied into a Word document and not uploaded as a script.

Abstract

1. The last sentence of the summary is not very informative - either provide a specific impact or drop the sentence.

Introduction

2. P3: “Students’ feelings of belonging may be especially important during academic transitions, such as the first year of college, when students are navigating new challenges for the first time.” I think this statement needs a reference.

3. P3, l. 54: Also, extraverts, seem to suffer more when they cannot pursue their inherent urge for social contact, see e.g., Weiß, M., Rodrigues, J., & Hewig, J. (2022). Big five personality factors in relation to coping with contact restrictions during the COVID-19 pandemic: A small sample study. Social Sciences, 11(10), 466.

4. P8, l. 175: Also, in terms of social media use, which is a way for most students to "belong.", see e.g., Weiß, M., Baumeister, H., Cohrdes, C., Deckert, J., Gründahl, M., Pryss, R., & Hein, G. (2022). Extraversion moderates the relationship between social media use and depression. Journal of Affective Disorders Reports, 8, 100343.

5. Overall, I like the introduction very much, but it could be streamlined to some extent. In the last paragraphs (p. 10), I think the authors mix research questions and hypotheses a bit. A clearer division (especially regarding the hypotheses) would be helpful.

Methods

6. P12 (participants): Wouldn't it be preferable for the interpretation of the results to include only those who answered all the questions? I understand that the numbers deviate heavily. Could the authors at least test whether the results would differ if only the students who answered all questions were analyzed?

7. P13: There are some very low reliabilities in these questionnaires. The authors should address this as a limitation.

8. P14: I think it's fine not to include the pre-registered exploratory analyses in the main paper. But why not in a supplement? Or are the results so interesting that the authors are planning a separate paper? I would like to see more transparency here.

Results

9. The subheadings (e.g., Is Students’ Personality at the Beginning of College Predictive of Their Belonging at the End of Their First Year?) read like undirected hypotheses, which is confusing (at least to me) – I would prefer statements. Also, I personally find the "yes" after each of these questions odd. You want to confirm or reject hypotheses, not have a dialogue with your subheadings.

10. Why did the authors not present any figures at all? I think some visualizations for the most important results would help the reader.

11. Could the authors provide test-retest reliability for the Big Five traits? With the given sample size this would be a nice additional information for interested readers.

6. PLOS authors have the option to publish the peer review history of their article (what does this mean?). If published, this will include your full peer review and any attached files.

Reviewer #1: No

Reviewer #2: No

---

## [Author Response · Author response to Decision Letter 0]

18 Nov 2023

We have comprehensively addressed all editor and reviewer comments, and attached a separate document titled "Response to reviewers" as part of our resubmission materials.

---

## [Decision Letter · Decision Letter 1]

22 Nov 2023

Who Feels Like They Belong? Personality and Belonging in College

PONE-D-23-21304R1

Dear Dr. Gopalan,

We’re pleased to inform you that your manuscript has been judged scientifically suitable for publication and will be formally accepted for publication once it meets all outstanding technical requirements.

Kind regards,

Sohaib Mustafa

Academic Editor

PLOS ONE

Additional Editor Comments (optional):

Reviewers' comments:

Reviewer's Responses to Questions

**Comments to the Author**

1. If the authors have adequately addressed your comments raised in a previous round of review and you feel that this manuscript is now acceptable for publication, you may indicate that here to bypass the “Comments to the Author” section, enter your conflict of interest statement in the “Confidential to Editor” section, and submit your "Accept" recommendation.

Reviewer #2: All comments have been addressed

2. Is the manuscript technically sound, and do the data support the conclusions?

Reviewer #2: Yes

3. Has the statistical analysis been performed appropriately and rigorously? 

Reviewer #2: Yes

4. Have the authors made all data underlying the findings in their manuscript fully available?

Reviewer #2: No

5. Is the manuscript presented in an intelligible fashion and written in standard English?

Reviewer #2: Yes

6. Review Comments to the Author

Reviewer #2: The authors did a great job with the revision and addressed my comments to my satisfaction. However, the data and scripts on OSF are only accessible with a requested access. I strongly encourage the authors to make them openly available without restrictions. In summary, I can recommend the paper for publication.

7. PLOS authors have the option to publish the peer review history of their article (what does this mean?). If published, this will include your full peer review and any attached files.

Reviewer #2: No

---

## [Editor Report · Acceptance letter]

1 Dec 2023

PONE-D-23-21304R1 

Who feels like they belong? Personality and belonging in college 

Dear Dr. Gopalan:

I'm pleased to inform you that your manuscript has been deemed suitable for publication in PLOS ONE. Congratulations! Your manuscript is now with our production department. 

Kind regards, 

on behalf of

Dr. SOHAIB MUSTAFA 

Academic Editor

PLOS ONE